# CXCR1/2 Inhibitor Ladarixin Ameliorates the Insulin Resistance of 3T3-L1 Adipocytes by Inhibiting Inflammation and Improving Insulin Signaling

**DOI:** 10.3390/cells10092324

**Published:** 2021-09-06

**Authors:** Vanessa Castelli, Laura Brandolini, Michele d’Angelo, Cristina Giorgio, Margherita Alfonsetti, Pasquale Cocchiaro, Francesca Lombardi, Annamaria Cimini, Marcello Allegretti

**Affiliations:** 1Department of Life, Health and Environmental Sciences, University of L’Aquila, 67100 L’Aquila, Italy; vanessa.castelli@univaq.it (V.C.); michele.dangelo@univaq.it (M.d.); margherita.alfonsetti@guest.univaq.it (M.A.); francesca.lombardi@univaq.it (F.L.); 2Dompè Farmaceutici SpA, Via Campo di Pile, 67100 L’Aquila, Italy; laura.brandolini@dompe.com (L.B.); cristina.giorgio@dompe.com (C.G.); pasquale.cocchiaro@dompe.com (P.C.); 3Sbarro Institute for Cancer Research and Molecular Medicine and Center for Biotechnology, Temple University, Philadelphia, PA 19122, USA

**Keywords:** diabetes, obesity, inflammation, glucose uptake, insulin resistance, pharmacological approach

## Abstract

Type 2 diabetes mellitus is a severe public health issue worldwide. It displays a harmful effect on different organs as the eyes, kidneys and neural cells due to insulin resistance and high blood glucose concentrations. To date, the available treatments for this disorder remain limited. Several reports have correlated obesity with type 2 diabetes. Mainly, dysfunctional adipocytes and the regulation of high secretion of inflammatory cytokines are the crucial links between obesity and insulin resistance. Several clinical and epidemiological studies have also correlated the onset of type 2 diabetes with inflammation, which is now indicated as a new target for type 2 diabetes treatment. Thus, it appears essential to discover new drugs able to inhibit the secretion of proinflammatory adipocytokines in type 2 diabetes. Adipocytes produce inflammatory cytokines in response to inflammation or high glucose levels. Once activated by a specific ligand, CXCR1 and CXCR2 mediate some cytokines’ effects by activating an intracellular signal cascade once activated by a specific ligand. Therefore, it is conceivable to hypothesize that a specific antagonist of these receptors may ameliorate type 2 diabetes and glucose metabolism. Herein, differentiated 3T3-L1-adipocytes were subjected to high glucose or inflammatory conditions or the combination of both and then treated with ladarixin, a CXCR1/2 inhibitor. The results obtained point towards the positive regulation by ladarixin on insulin sensitivity, glucose transporters GLUT1 and GLUT4, cytokine proteome profile and lipid metabolism, thus suggesting ladarixin as a potentially helpful treatment in type 2 diabetes mellitus and obesity.

## 1. Introduction

Obesity is related with an increased risk of insulin resistance and several metabolic complications, including type 2 diabetes mellitus (T2D) [1,2]. The prevalence of obesity and obesity-related diseases including type 2 diabetes, cardiovascular disorders and metabolic syndrome has significantly risen in the past 20 to 30 years [3]. Adipose tissue (AT) plays an essential role in the development of these disorders due to adipocyte dysfunction and altered secretion of adipokine occurring in obesity conditions [4,5]. The primary function of AT is to stimulate the uptake of both glucose and fatty acids after feed in response to insulin signalling [6]. These functions are impaired during the initiation and progression of obesity. At the beginning of obesity condition, an enlargement of the adipocyte accumulation arises, initially through an increase of individual adipocytes size due to lipids accumulation (cellular hypertrophy) [7,8]. As obesity progresses, adipocytes increase in number (hyperplasia) and their metabolic activity is dramatically altered [8]. The insulin-dependent glucose uptake in fat cells is dependant by glucose transporter type 4 (GLUT4, insulin-responsive) and exerts a crucial part in glucose homeostasis of the entire body [9]. GLUT4 is upregulated in differentiated adipocytes and matures as GLUT1 (glucose transporter type 1, non-insulin-responsive) is downregulated [10,11]. Under obesogenic circumstances, GLUT4 expression is significantly decreased in AT [12]. Decreased GLUT4 levels and decreased insulin-dependent glucose uptake are main characteristics of adipocyte impairment in hypertrophic AT [13]. A second crucial role of healthy adipocytes is the generation and release of adipokines, including adiponectin [4]. Adiponectin expression raises 3 to 4 days post-differentiation and is believed to be a late marker of mature adipocytes [14,15]. Comparable to GLUT4, the expression of adiponectin is considerably decreased in obesity, consequently acting as an additional biomarker of impaired adipocyte function [16].

Furthermore, obesity, considered as a condition of low-level inflammation, is associated with insulin resistance. In AT of obese patients, recruited macrophages can trigger the inflammatory pathway activation in local adipocytes by secreting pro-inflammatory cytokines, i.e., tumor necrosis factor alpha (TNF-α) and interleukin-6 (IL-6) [17]. These inflammatory activation impairs insulin receptor substrate-1 (IRS-1) function and downstream insulin/PI3K pathway to prevent glucose uptake, inducing to insulin resistance in adipocytes [18].

In particular, it has been shown that IL-8 can be secreted by human adipose tissue and/or adipocytes, indicating that IL-8 is involved in some obesity-related health complications [19], such as insulin resistance (IR), atherosclerosis, cardiovascular disorders. IL-8, acting through CXCR1/2, is the main adipocytokine that causes IR through the inhibition of insulin-induced Akt phosphorylation in adipocytes [20]. Indeed, serum levels of IL-8 are increased in T2D patients [21], and CXCR2 (−/−) mice are resistant to diet-induced insulin resistance and diabetes [22]. Previous studies reported an increase in the concentration of CXCL1/KC (homologous to human IL-8) in plasma and adipose tissue of obese mouse models (ob/ob and high-fat diet) [23]. Furthermore, bone marrow chimera mice deficient in the KC receptor CXCR2 showed a decrease in obesity-induced inflammation and insulin resistance compared to controls [23]. A relationship between CXCR1/2 chemokine receptors and the advancement of insulitis and diabetes in mice has been previously established, demonstrating that transient CXCR1/2 inhibition inhibits inflammation-mediated islet impairment in the mild streptozotocin models [24]. Thus, the attenuation of IL-8 action, which acts through CXCR1/2, may be proposed as a target for the prevention or treatment of diabetes and its related disorders [20].

Ladarixin, a dual inhibitor of IL-8 receptors CXCR1 and CXCR2 with an optimal pharmacokinetic profile [25] that has completed phase I studies and already entered phase II/III trials, was previously used in in vivo murine model to achieve the pharmacologic inhibition of the CXCL1–CXCR1/2 axis [24], obtaining promising results in maintaining residual B-cells and making the potential to significantly adjust the approach for managing human type 1 diabetes. In addition, Ladarixin emerged as an useful candidate to treat a wide range of neutrophilic-mediated respiratory disorders since it has been demonstrated to strongly reduce the release of the pro-inflammatory cytokine [26].

Based on the exposed evidence, the present work aims to investigate the potential effects of ladarixin in 3T3-L1 adipocytes upon inflammatory and diabetic conditions. We also explored the molecular mechanisms underlying the anti-diabetic and anti-inflammatory effects.

## 2. Materials and Methods

### 2.1. Cell Cultures

Mouse preadipocytes 3T3-L1 cell line was purchased from ATCC (American Type Culture Collection, #CL-173, Lot. 70028180) and cultured in Dulbecco’s modified Eagle Medium (ATCC, #30-2002) supplemented with non-heat-inactivated 10% Bovine Calf serum (ATCC, USA) at 37 °C in a 5% CO_2_ atmosphere until confluence. At 2 days after full confluence, cells were differentiated via incubation in DMEM containing 10% fetal bovine serum (FBS), 0.5 mM isobutylmethylxanthine (IBMX), 1 μM dexamethasone, 1 μg/mL insulin for 48 h and then for 2 days in DMEM (10% FBS) containing 10 μg/mL insulin alone (Sigma-Aldrich, St. Louis, MO, USA). After 22 days, over 80% of the cells exhibited the adipocyte phenotype with large lipid droplets in the cytoplasm and by the expression of adipocytic markers, such as Pre-adipocyte factor-1 (Pref-1), adipose tissue fatty acid-binding protein (aP2) and peroxisome proliferator-activated receptor-gamma (PPARγ). Cells were maintained and refed every 2 days with DMEM 10% FBS 25 mM glucose (for Normal Glucose-NG condition) or 50 mM (25 mM added over the 25 mM concentration already present in the DMEM culture media) for High Glucose condition (HG). Once the model was established, adipocytes were subjected to inflammatory challenge (INF) by using the conditioned medium of activated macrophage or by using KC (mouse homologous to human IL-8). For the experimental set up the passage number used was 3.

Murine macrophages RAW 264.7 cell line was purchased from ATCC (#TIB-71, Lot. 70026471) and cultured with Dulbecco’s modified Eagle Medium (ATCC, #30-2002) supplemented with FBS 10%. Macrophages RAW 264.7 were exposed to 0.1 mg/mL lipopolysaccharide (LPS, from Sigma-Aldrich, USA) for 24 h then the conditioned medium was collected, filtered and used for INF conditions. For the experimental set up the passage number used was 2.

KC stimulus was obtained by adding 20 ng/mL of cytokine mouse KC (Miltenyi Biotec, Cologne, Germany to the media for 24 h as previously reported by [23,27]. Ladarixin (DF2156A, PubChem CID 23709380) was freshly prepared in incomplete media at the concentration of 25 mM; the subsequent dilutions were prepared in culture media.

Specifically, ladarixin was added 24 h after stimuli (after KC or INF challenge cells were gently washed with PBS) and incubated for 72 h. Dose-response effects of Ladarixin (1–50 µM) were evaluated basing on cell viability (MTS assay and Cytotox assay), GLUT4 protein levels (GLUT4 ELISA and WB) and CXCR2 expressions (RT-PCR). Based on the results obtained, the concentration 10 µM was selected for all subsequent experiments.

### 2.2. Treatments

Cells were divided into 12 groups:Controls cells (NG),High glucose cells (HG),NG plus inflammatory stimulus (NG-INF),HG plus inflammatory stimulus (HG-INF),NG plus KC (NG-KC),HG plus KC (HG-KC),NG-treated with ladarixin (NG-LAD),HG-treated with ladarixin (HG-LAD),NG-INF-treated with ladarixin (NG-INF-LAD),HG-INF-treated with ladarixin (HG-INF-LAD),NG plus KC with ladarixin (NG-KC-LAD),HG plus KC with ladarixin (HG-KC-LAD).

### 2.3. Nile Red Staining

Nile Red is a widely used neutral lipid stain that helps detect the adipogenesis of various types of cultured cells. Briefly, after culturing as previously described for 22 days, cells were washed with PBS and incubated with Nile Red (final concentration 50 ng/mL) for 30 min, then washed again with PBS. Undifferentiated and differentiated 3T3-L1 cells stained with Nile red were observed at fluorescence microscopy (Olympus, Leica, Wetzlar, Germany).

### 2.4. Western Blotting

Control and treated cells were collected and lysed in ice-cold RIPA buffer containing protease and phosphatase inhibitor cocktails (Sigma-Aldrich). The extracted proteins were quantified using BCA assay (Thermo Fisher Scientific, Waltham, MA, USA). Protein lysates (30 μg) were diluted in sample buffer and denaturing agent of Invitrogen (Thermo Fisher Scientific) and then heated at 70 °C for 10 min in thermoblock (Eppendorf, Hamburg, Germany).The samples analyzed for GLUT4 were not heat-denatured. Samples were separated on gradient Nupage Bis-Tris Glycine gradient precast gel (Invitrogen, USA) and electroblotted onto a polyvinyl difluoride membrane (PVDF, Millipore, Darmstadt, Germany using a semi-dry system (Thermo, USA). Non-specific binding sites were blocked by Blocking Buffer (Invitrogen, USA) for 10 min at RT. Membranes were then incubated overnight at 4 °C with the following primary antibodies, diluted in blocking buffer: mouse anti-tubulin alpha 1:2000 (ab80779, lot: GR209729-4, Abcam, Cambrige, UK); mouse anti-AP2 1:200 (MA1-872, lot:UL296310, Invitrogen, USA), rabbit anti-PPARγ 1:1000 (ab209350, lot:GR3284934-2, Abcam, Cambridge, UK), rabbit anti-Pref1 1:200 (MBS9210091, LOT: SA141106AX, MyBioSource, San Diego, CA, USA), rabbit anti-GLUT4 1:1000 (PA5-23052, lot: PH1895639, Thermo, USA), rabbit anti-NfkB 1:1000 (ab16502, lot:GR3309481-1, Abcam, USA), rabbit anti-p-IRS1 1:500 (2385S, lot:2, Cell Signaling, Danvers, MA, USA), rabbit anti-p-IRS2 1:500 (4502S, lot:6, Cell Signaling, USA), rabbit anti-IRS 1:500 (3407S, lot:6, Cell Signaling, USA) rabbit anti-PI3k 1:1000 (ab151549, lot:GR3235734, Abcam, USA), rabbit anti-p-AKT (ab81283, lot:GR240003, Abcam, USA) and rabbit anti-AKT 1:1000 (9272S, lot:2, Cell Signaling, USA), mouse conjugated Actin 1:20000 (5125S, lot:6, Cell Signaling, USA). As secondary antibodies, 1:20,000 peroxidase-conjugated anti-rabbit (111-035-003, lot:149393, Jackson Immuno Research, Ely, UK) or anti-mouse IgG (115-035-003, lot:149723, Jackson Immuno Research, Cambridge, UK) were used. Immunoreactive bands were visualized by chemiluminescent substrate (Thermo, USA), according to the manufacturer’s instructions, and visualized at Uvtec digital system (Cambridge, UK). The relative densities of the immunoreactive bands were determined and normalized to tubulin or actin or their respective total forms using Fiji software. To normalize the GLUT4 plasma membrane Coomassie staining was used. Values were given as relative units.

### 2.5. Incucyte Cytotox Green Assay

3T3-L1 were plated (seeding density 1 × 10^5^ cells/cm^2^) into a 24-wells plate and differentiated as described above. Then, the cells were exposed to ladarixin treatments (10–25 µM) and different concentrations of FBS, and 250 nM of IncuCyte Cytotox Green Reagent (Essen BioScience, Newark, UK) were added in the experimental culture medium for counting dead cells. The plates were put in IncuCyte device (20 × objectives). The cytotoxicity activation was recorded (three images for well) every 3 h by both fluorescence scanning and phase contrast for 72 h at 37 °C and 5% CO_2_. Images were analyzed utilizing the Incucyte ZOOM software (Newark, UK), and the data were reported as green object count (per image).

### 2.6. GLUT4 ELISA Kit

The amount of GLUT4 in adipocyte plasma membranes was evaluated using the ELISA KIT by MyBioSource (#MBS2022770) following the manufacturer’s instructions. Briefly, the cells were counted and lysed as suggested, then the Standard curve and the reagents were prepared and used following the assay procedure. The microplate supplied in this kit has been pre-coated with an antibody specific to GLUT4. Standards or samples are then added to the proper microplate wells with a biotin-conjugated antibody specific to GLUT4. Then, avidin conjugated to HRP is added to each microplate well and incubated. Following the addition of TMB substrate solution, only those wells that contain GLUT4, biotin-conjugated antibody and enzyme-conjugated avidin will display a color change. The addition of sulphuric acid solution terminates the enzyme-substrate reaction, and the color shift is measured using a microplate reader at a wavelength of 450 nm ± 10 nm (Spark, Tecan, Mannedorf, Switzerland). The concentration of GLUT4 in the samples is then revealed by comparing the O.D. of the samples to the standard curve. Data were presented as ng/mL.

### 2.7. Real-Time PCR

Total RNA was extracted by Trizol reagent (Thermo), according to the manufacturer’s instructions after 24 h post-treatment. The total RNA concentration was determined in RNAase-free water using Nanodrop, while the concentration was determined using Qubit Fluorometer 3.0 (Thermo). 1 μg of total RNA was reverse transcribed using a 5X All-In-One RT MasterMix (Applied Biological Materials, Richmond, BC, Canada) into cDNA using thermo-block (Eppendorf). Finally, the real-time PCR was carried out on ABI 7300HT sequence detection system (ABI), containing 2X TaqMan Gene Expression Master Mix (Invitrogen, USA), DEPC water and 5 μL of cDNA and 1 μL the following primers: Prime Time qPCR Assay: GLUT4, CXCR2, IL-10, CXCL-1, TNFA, GLUT1 and GLUT4 qRnoCIP0027857 were purchased from Biorad, Des Plaines, IL, USA. Triplicate samples were run for each gene. The reference gene GAPDH qRnoCIP0050838 was applied as an internal control to normalize the expression of target genes. Relative expression levels were determined for each sample after normalization against the reference gene, using the ΔΔCt method to compare relative fold expression differences.

### 2.8. Lipolysis Colorimetric Assay Kit

To evaluate the lipolysis, cells were grown and differentiated as previously described in a 48 wells plate. After 22 days, cells were washed two times with wash buffer, and the wash buffer was replaced with lipolysis assay buffer. Then, isoproterenol (100 nM) was added to stimulate lipolysis for 2 h and the medium was collected. 25 µL of media were added into 96 well-plates, and the volume was adjusted with lipolysis Assay buffer to 50 µL. The standard curve was prepared following the manufacturer’s protocol as well as the reaction mix. The plate was incubated at room temperature for 30 min protected from light. The results were read at OD 570 nm in a plate reader. Data were expressed as glycerol content (nmol/well).

### 2.9. Glucose Uptzake

To monitor the uptake of glucose, mature adipocyte 3T3-L1 (plated in a 48 wells plate as described above) were incubated with 1 mM of the fluorescent tracer 2-NBDG (2-Deoxy-2-[(7-nitro-2,1,3-benzoxadiazol-4-yl)amino]-d-glucose; Sigma, USA) for 10 min at room temperature (after a gentle washing). The fluorescence intensity was measured at Ex/Em = 485/535 nm. Data are expressed as 2-NBDG fluorescence intensity (% NG control).

### 2.10. KC ELISA

To evaluate the release of KC from mature adipocytes upon different conditions the plate was prepared the day before and analyzed using Capture Antibody, following manufacturer’s protocol (R&D System, Minneapolis, MN, USA). Then, the assay procedure was followed by preparing the standard curve and the samples, then the substrate solution was incubated for 20 min at room temperature, followed by the addition of stop solution. The optical density of each well was immediately determined using a microplate reader set to 450 nm. Data were expressed as pg/mL.

### 2.11. Plasma Membrane Protein Extraction

To evaluate the amount of GLUT4 in adipocyte plasma membranes, the plasma membrane protein extraction kit was used following the manufacturer’s instructions (Abcam, UK). This kit provides optimized buffers and reagents to extract plasma membrane proteins from mammalian tissues and cells. Briefly, cells were scraped in PBS and then spun down (3000 rpm for 5 min). After a wash in ice-cold PBS, the cells were re-suspended in 2 mL of the Homogenize Buffer Mix in an ice-cold Dounce homogenizer. Cells were then homogenized on ice 30–50 times. The homogenate was transferred to multiple 1.5 mL microcentrifuge tubes and centrifuged in 700× *g* for 10 min at +4 °C. The supernatant was collected and moved to new vials and centrifuged at 10,000× *g* for 30 min at +4 °C. Finally, the pellet is the total cellular membrane protein (comprising proteins from both plasma membrane and cellular organelle membrane); thus, purification of plasma membrane proteins was performed, and the samples obtained were kept at −80 °C until use. The procedure offers consistent yield and high purity (over 90%).

### 2.12. Adiponectin Mouse ELISA Kit

To evaluate the released adiponectin in cell culture media from mature adipocytes at the different conditions tested, adiponectin mouse ELISA kit was performed. Briefly, the adiponectin standard curve was prepared and added to the plate and samples for 2 h. After extensive washes, a biotinylated adiponectin antibody was added to each well and incubated for one hour. After other washes, the 1X SP Conjugate was added to each well and incubated for 30 min at room temperature. After extensive washes, chromogen substrate was added for 12 min, followed by the addition of stop solution. The results were read immediately at 450 nm in a plate reader. Data were expressed as ng/mL.

### 2.13. Leptin Mouse ELISA Kit

To evaluate the released leptin from mature adipocytes at the different conditions tested, a leptin mouse ELISA kit was performed in cell culture media. Briefly, the leptin standard curve was prepared and added to the plate as well as samples for 2.5 h at room temperature. After extensive washes, a biotinylated leptin detection antibody was added to each well and incubated for one hour. After other washes, the 1X HRP- Streptavidin solution was added to each well and incubated for 45 min at room temperature. After extensive washes, a one-step substrate reagent was added for 30 min, followed by a stop solution. The results were read immediately at 450 nm in a plate reader. Data were expressed as pg/mL.

### 2.14. NFkB p65 (pS536) ELISA Kit

Cells were cultured, differentiated and treated as previously described. Samples were extracted and lysed as manufacturer’s protocol using chilled 1× cell extraction buffer PTR and assayed immediately. The protein concentration was evaluated and adjusted. The control lysate was prepared as well and added in the wells as well as samples and antibody cocktails for 1 h at room temperature. After extensive washes, TMB substrate was incubated for 15 min, and the stop solution was added for 1 min. The results were read immediately at 450 nm in a plate reader. Data were expressed as µg/mL.

### 2.15. Mouse Cytokine Array

Conditioned media was centrifugated to remove particulates and assayed immediately. The sample amount will be adjusted as suggested (500 µL). The reagents were prepared following the manufacturer’s protocols (R&D, USA). Briefly, the membranes were incubated with Array Buffer 6 for 1 h on a platform shaker. The samples were prepared by putting up to 1 mL of Array Buffer 4 in two separated tubes and then 15 µL of reconstituted mouse cytokine detection antibody cocktail (>40 mouse cytokines) to each prepared sample and incubated for 1 h. Then, the Array Buffer 6 was aspirated and replaced with the sample/antibody mixtures and incubated at 4 °C overnight. The following day, membranes were washed 3 times and Streptavidin-HRP (1:2000) was incubated for 30 min at room temperature on a platform shaker. The membranes were washed again, and 1 mL of prepared Chemi Reagent mix will be placed onto each membrane. Multiple exposure times will be acquired using UVITEC digital analyzer (Alliance, Cambridge, UK). The positive signals seen in the developed membranes can be detected by putting the transparency overlay template on the array image and aligning it with pairs of reference spots in the three corners of each array. Reference spots were included to reveal that the array is incubated with Streptavidin-HRP during the assay procedure. Pixel densities (average signals of pair of duplicates which represent each cytokine) were analyzed by ImageJ and the average background was subtracted from each spot.

### 2.16. Human Adipocytes Extraction from Patient Adipose Tissue

The adipose tissue was collected from “Ospedale San Salvatore”, L’Aquila and the patient provided written informed consent. The liposuction was completed under peripheral blocks achieved by using hyperbaric bupivacaine (10 mg) in epidural space. After the surgical area was set up, tissues were infiltrated with Klein solution (lidocaine/epinephrine) respecting the ratio of 1:1 (wet liposuction).

Through a 4 mm skin incision, the solution was administered with slow fan shape movements in fat thickness using a 15 cm blunt cannula (1 mm) designed for infiltration with 3 holes per side along its length. The surgical procedure was conducted through the same surgical access using a 2 mm blunt cannula connected to an electric negative pressure generator. Liposuction was performed with slow fan shape movements starting from the deeper layer moving up to the subcutaneous space. Immediately after surgery, the lipoaspirate was put in a sterile box and sent to the laboratory and processed by spontaneous stratification and centrifugation methods as previously described [28]. Briefly, sample aliquots of lipoaspirate were left at room temperature to acquire the sample stratification under the gravity effect. Aliquots then were subjected to different centrifugation for 3 min at room temperature. After these procedures, four layers were detected: free released oils (“oily fraction”) on the top, a “middle layer” consisting of purified fat with adipocytes and connective tissue, an aqueous layer containing Klein solution, and a bottom layer composed by erythrocytes and stromal cells. To obtain the adipocytes, the “middle layer”, collected in sterile conditions, was incubated with 1.5 mg/mL crude collagenase type I in phosphate buffer solution at 37 °C for 45 min in a water bath by gentle stirring. The collagenase activity was then neutralized by adding DMEM supplemented with 10% FCS and the digested tissue was centrifuged at 90× *g* for 3 min to recover adipocytes. The adipocytes obtained were cultured for 3 days in DMEM supplemented with 5% FBS, 1% BSA and 1% antibiotics and then centrifuged for 10 min at 80× *g*. The procedure was repeated, and cells were plated in a T75 flask. After 24 h cells were treated with Ladarixin in LAD condition as previously described for 72 h. Then images of human adipocytes were captured using optical microscopy (Leica, Wetzlar, Germania) and the adipocyte diameter was measured using Fiji comparing the dimension to the scale bar, then the mean of values was reported in the graph and expressed as µm. The media were collected and gently centrifuged to remove cell debris and freshly used for cytokine Array Kit procedure.

### 2.17. Real-Time PCR for Human Adipocytes

Total RNA was extracted by Trizol reagent (Thermo, USA), according to the manufacturer’s instructions after 24 h post-treatment. To improve the extraction procedure, detergents (Deoxycholate-Nonidet P40) were added. The total RNA concentration was determined in RNAase-free water using Nanodrop, while the concentration was determined using Qubit Fluorometer 3.0 (Thermo, USA). 1 μg of total RNA was reverse transcribed using a 5X All-In-One RT MasterMix (Applied Biological Materials, Canada) into cDNA using Thermo-block (Eppendorf). Finally, the real-time PCR was carried out on ABI 7300HT sequence detection system (ABI), containing 2X TaqMan Gene Expression Master Mix (Invitrogen, USA), DEPC water and 5 μL of cDNA and 1 μL of the following primers. Prime Time qPCR Assay: human CXCL8, CXCL6, GLUT4 were purchased from Biorad, USA. Triplicate samples were run for each gene. The reference gene GAPDH was used as an internal control to normalize the expression of target genes. Relative expression levels were analyzed for each sample after normalization against the reference gene, using the ΔΔCt method for comparing relative fold expression differences.

### 2.18. Human XL Cytokine Array Kit

Conditioned media was centrifugated to remove particulates and assayed immediately. The sample amount will be adjusted as suggested (500 µL). The reagents were prepared following the manufacturer’s protocols (R&D, USA). Briefly, the membranes were incubated with Array Buffer 6 for 1 h on a rocking platform shaker. The samples were prepared by adding up to 1 mL of Array Buffer 4 in two separated tubes and incubated for 1 h. Then, the Array Buffer 6 was aspirated and replaced with the sample/antibody mixtures and incubated at 4 °C overnight. The following day, membranes were washed 3 times. For each array, 30 µL of Detection Antibody Cocktail were added to 1.5 of 1× array Buffer 6 and pipetted in the wells and incubated for 1h at RT. After extensive washes, Streptavidin-HRP (1:2000) was incubated for 30 min at room temperature on a rocking platform shaker. The membranes were washed again, and 1 mL of prepared Chemi Reagent mix will be placed onto each membrane. Multiple exposure times will be acquired using UVITEC digital analyzer (Alliance, Cambridge, UK). The positive signals seen in the developed membranes can be identified by placing the transparency overlay template on the array image and aligning it with pairs of reference spots in the three corners of each array. Reference spots were included to demonstrate that the array is incubated with Streptavidin-HRP during the assay procedure. Pixel densities (average signals of pair of duplicates which represent each cytokine) were analyzed by ImageJ and the average background was subtracted from each spot.

### 2.19. Statistical Analyses

All data were presented as mean ± SE. Data analyses were performed using GraphPad Prism 8 (GraphPad Software Inc., San Diego, CA, USA). The significance of differences between the two groups was determined by *t*-test analyses, while for multiple comparisons one-way analysis of variance (ANOVA) followed by Tukey post-hoc tests was used. The level of significance was set at *p* < 0.05.

## 3. Results

### 3.1. Adipocyte Model Development

3T3-L1 cells were chemically differentiated from fibroblasts to mature adipocytes as described in the Methods section. Contrast phase microscopy was used to follow the formation of lipid droplets (Figure 1A). 3T3-L1 were stained before induction and after 22 days with Nile red (Figure 1B). At 22 days more than 80% of cells appeared fully differentiated toward mature adipocytes, showing large lipid droplets condition (Figure 1A). Nile red staining confirmed complete differentiation (Figure 1B).

Adipocyte characterization was also corroborated by Western blotting analysis of pre-adipocyte factor 1 (Pref-1), adipose tissue fatty acid-binding protein (aP2), peroxisome proliferator-activated receptor-gamma (PPARγ) and GLUT4. aP2 is a carrier protein for fatty acids primarily expressed in adipocytes and macrophages. It is a crucial mediator of intracellular transport and the metabolism of fatty acids. Its expression during adipocyte differentiation is regulated through the action of PPARγ [29]. In line with these findings, protein levels of aP2 and PPARγ were found increased in differentiated compared to undifferentiated 3T3-L1 cells. Notably, the levels of GLUT4 resulted increased in differentiated adipocytes compared to pre-adipocytes (Figure 1C).

Once the model was established, cells were treated with different concentrations of ladarixin (1–50 µM) for 72 h and cells viability was evaluated by Cytotox Green Assay; the results showed that the concentrations tested were not toxic (Figure A1A,B).

### 3.2. CXCR1/2 Expression in Differentiated 3T3-L1 Cells and Ladarixin Dose-Range Findings

In the first set of experiments, CXCR2 expression was evaluated in NG and HG conditions by RT-PCR. Ladarixin was incubated at 1–50 µM for 24 h. In NG conditions, ladarixin did not affect CXCR2 gene expression, whereas in HG conditions CXCR2 levels in control (CTR) cells were significantly increased and ladarixin strongly counteracted this effect starting from 10 µM (Figure 2A).

Protein levels of GLUT4, the primary glucose transporter responsible for insulin-stimulated glucose uptake in AT, were also evaluated in NG and HG conditions by ELISA and Western blotting analyses. Ladarixin was incubated at 1–50 µM for 72 h. In HG conditions, GLUT4 levels were significantly decreased compared to control cells in NG conditions (Figure 2B,C) and this effect was reverted by Ladarixin treatment starting from 10 µM (Figure 2B,C). Based on these results, the 10 µM concentration of ladarixin was used for the subsequent experiments.

### 3.3. Effects of Ladarixin Treatment on HG-Induced and Inflammation-Induced Activation of Inflammatory Signals in Differentiated 3T3-L1 Cells

Both HG condition and INF stimulation strongly increased KC expression as determined in culture supernatants by ELISA assay. The exposure to INF in differentiated 3T3-L1 cells culture in the HG condition did not result in further amplification of KC production. A similar effect was observed by stimulating cells with KC suggesting a direct role of CXCR2 in the induction of ligand overexpression. Consistently, ladarixin significantly decreased KC levels (pg/mL) in all tested conditions (Figure 3A).

Levels of the active form of NFKB (phosphorylation in S536) were also evaluated by ELISA assay. Interestingly, in all HG conditions and upon inflammatory or KC stimuli and in NG conditions, NFKB phosphorylated form (µg/mL) significantly increased, while ladarixin treatment restored control conditions (Figure 3B). These data were also confirmed by Western blotting analysis for NFKB nuclear fraction (Figure 3C).

The effect of HG and INF on Cxcl1 was also evaluated by RT-PCR at 24 h (Figure 4A) confirming a moderate increase of the chemokine expression induced by INF and KC in normoglycemic conditions and a marked amplification induced by high glucose concentration. The ability of ladarixin to contrast Cxcl1 amplification in all the tested conditions remarked a key role of CXCR2. Similar behavior was observed when IL-6, Tnfα and IL-10 gene expressions were evaluated by RT-PCR at 24 h (Figure 4B–D). In HG conditions, the pro-inflammatory cytokines IL-6 and Tnfα were significantly increased whereas a marked reduction of the anti-inflammatory cytokine IL-10 gene expression was observed. As observed for Cxcl1, INF and KC stimuli alone caused a similar trend but to a lower extent. Ladarixin treatment completely reverted the effects of HG exposure in the presence or absence of additive inflammatory activation (Figure 4B–D).

Finally, we investigated the cytokine proteome profile in 3T3-L1 cells upon inflammatory stimuli and/or HG conditions with and without ladarixin treatment (Figure 5 and Figure A2). KC insult confirmed to exert a selective induction of KC secretion with a moderate general increase of other inflammatory mediators. HG and INF (in a larger extent) conditions induced the secretion of a large panel of pro-inflammatory cytokines in mature adipocytes. Together with KC, the cytokines TIMP-1, TNFα and IFN-**γ****,** and the chemokines MCP-1 and BCA-1, key players in chronic systemic inflammation occurring in diabetes and obesity, were the most abundant. 

Interestingly, ladarixin exhibited a broad inhibitory effect on the secretion of the analyzed inflammatory mediators, not only in response to KC stimulations but also under the HG and INF conditions (Figure 5 and Figure A2), suggesting a pivotal role of the CXCR1/2 axis in the secretion of inflammatory mediators by mature adipocytes under stressed condition.

### 3.4. Effects of Ladarixin Treatment on Primary Human Adipocytes

Finally, the effects of ladarixin were evaluated in the primary culture of human adipocytes from an obese patient. The phase-contrast microscopy indicated that in obese conditions, adipocytes appear as large cells full of lipids, while upon ladarixin, the size of the cells appears strongly reduced, thus indicating a lipid mobilization (Figure 6A,B). Then, the RT-PCR for CXCL8, CXCL6 and GLUT4 was performed. Notably, ladarixin treatment was able to significantly reduce CXCL8, CXCL6 expression as well as GLUT4 levels (Figure 6C). The secretome profile analysis indicates a strong presence of almost all the protein assayed, among that adiponectin, leptin, the cytokines IL-6, IL-8, GROα, ENA-78 and the angiogenic factors angiogenin and VEGF appear strongly reduced by ladarixin treatment, thus suggesting that ladarixin treatment was able to counteract the metabolic impairment in the obese patient (Figure 6D). Whereas the inhibitory effect on key inflammatory pathways confirms the findings obtained in murine adipocytes, the marked reduction of leptin levels suggests a different role of CXCR2 in human cells in the regulation of hormone release.

### 3.5. Effects of Ladarixin Treatment on HG-Induced and Inflammation-Induced Regulation of Glucose Transport and Insulin Response in Differentiated 3T3-L1 Cells

Having found that HG dramatically downregulated GLUT4 expression and that this effect was reverted by Ladarixin treatment, we investigated the effect of HG on the expression of GLUT1 which, together with GLUT4, constitutes one of the two predominant isoforms of the glucose transporter expressed in 3T3-L1 adipocytes. Despite there was not a marked increase in GLUT1 in all normoglycemic conditions and HG alone or presence of INF and KC, the treatment with Ladarixin in HG in presence of both INF and KC strongly increased GLUT1 expression (Figure 7A), thus suggesting a specific role of CXCR2 in inhibiting the inflammation-induced upregulation of this transporter in hyperglycemic conditions. A similar trend was observed when the expression of GLUT4 was measured under NG and HG conditions with or without INF and KC stimuli (Figure 7B).

IR was measured as the consumption of 2-NBDG and the release of glycerol. In particular, the uptake of glucose was measured in mature 3T3-L1 adipocytes by fluorescent tracer 2-NBDG. In NG conditions, no differences were observed among the different treatment groups, while in HG conditions, glucose uptake was significantly decreased both in control and under inflammatory or KC stimuli; ladarixin treatment, consistently with the observed effects on glucose transporters, counteracted this effect and both concentrations tested (50 and 10 µM) (Figure 7C).

The lipolysis process was evaluated using the ELISA assay for glycerol release. In NG conditions, ladarixin treatment was found to inhibit both the basal release and the incremental release induced by INF and KC at both the concentration tested. In HG conditions, a marked increase in glycerol release was observed, and no additive effect was observed in presence of the inflammatory stimuli. Nevertheless, ladarixin treatment, while reducing by 50% the HG-induced lipolysis effect, showed a modest inhibition when inflammatory stimuli were added to the HG medium (Figure 7D). Higher concentrations of ladarixin may be needed for effective inhibition of the process but it is plausible that the combination of the HG and INF stimuli, while not affecting the overall glycerol release, activate redundant pathways.

Secretion of major adipokines, adiponectin and leptin, which can affect inflammation and insulin sensitivity, were analyzed by ELISA assay (Figure 8A,B).

Regarding adiponectin, no significant effects of inflammatory stimuli and treatment with the CXCR2 inhibitor were observed in normoglycemic conditions. In contrast, in HG conditions, both in the control group and the INF and KC groups showed reduced adiponectin levels compared to NG groups. Ladarixin treatment was able to determine a significant increase of the adiponectin levels, suggesting a restoration of the metabolic activity (Figure 8A).

Differently, the levels of leptin, an important regulator of lipid metabolism, were not significantly affected by the inflammatory stimuli in normoglycemic conditions, CXCR2 inhibition by ladarixin treatment resulted in a stimulation of leptin secretion under basal condition, an effect that was maintained under inflammatory stimuli (Figure 8B). In HG conditions, inflammatory stimuli caused a marked increase in leptin release. Ladarixin significantly increased the level of leptin in the control group whereas an incremental increase was not observed in presence of inflammatory stimulation (Figure 8B).

CXCR2 activation has been previously reported to inhibit insulin-induced AKT phosphorylation in muscle cells and fibroblasts [30]. The PI3K/AKT signaling pathway is implicated in crucial biological processes involving cell proliferation, differentiation, metabolism and cytoskeletal reorganization. Alterations to PI3K/AKT cascade are associated with metabolic diseases, such as obesity, diabetes and associated with insulin resistance [31,32].

To examine whether the PI3K/Akt pathway was responsible for the ladarixin-mediated ameliorations in glucose uptake and insulin resistance, the PI3K p110 and pAkt (Ser 473) were analyzed. Interestingly, we observed a decrease in PI3K p110 and AKT phosphorylation in HG conditions with or without combined inflammatory or KC stimuli (Figure 9A). Inhibition of CXCR2 by ladarixin restored PI3K p110 and AKT phosphorylation (Ser473) in hyperglycemic conditions only when given in concomitance with INF and KC stimuli (increased levels), confirming a role of CXCR2 activation in the suppression of the PI3K/AKT pathway (Figure 9A). In particular, the activation of the subunit p110 is sufficient for GLUT4 translocation [33]. Our results suggests that ladarixin stimulates glucose uptake acting through PI3K/Akt pathway.

A tendency to increase AKT phosphorylation under inflammatory stimulation was also observed in normoglycemic conditions (Figure 9A). 

To evaluate the effect of CXCR2 inhibition on insulin signalling in murine adipocytes under stressed conditions, we analyzed the phosphorylation of the insulin receptor substrates IRS-1 (Ser1101) and IRS-2 (Ser388) by Western blotting. 

Insulin stimulation of glucose transport in adipocytes requires insulin receptor-mediated tyrosine phosphorylation of IRS -1 and -2 [34]. In line with the hypothesis, phosphorylation of IRS-1 and IRS-2 resulted increased upon ladarixin treatment in HG plus inflammatory or KC stimulus, same conditions in which an effect on GLUT4 and GLUT1 expression was previously described, indicating the ability of CXCR2 inhibition to restore insulin-mediated metabolic activity (Figure 9B). By contrast, an opposite effect was observed in the normoglycemic condition in response to INF and KC stimuli, as well as in HG condition alone, indicating a fine-tuning role of CXCR2 on insulin receptor activation in murine adipocytes that ranges from activation to inhibition according to the specific conditions. 

## 4. Discussion

AT inflammation is a crucial condition for the development of the obesity-related disorders. NF-κB, a transcription factor that controls numerous pro-inflammatory proteins production (including TNF-α, IL-6 and IL- 1β), has been recognized to be crucial in the development of diabetes and obesity-related diseases. In cell culture and animal studies, increased NF-κB activity was related with insulin resistance and muscle atrophy [35,36] and with worsening the homeostasis of lipid and glucose metabolism in AT [6]. Circulating plasma levels of several immuno-modulatory peptides, including TNFα, IL-6, IL-8, IL-10, IL-18 and C-reactive protein, were found to be elevated in human obesity [37]. In obesity and obesity-related disorders, such as diabetes, the key actors in the systemic chronic inflammation are the incrementing of pro-inflammatory macrophages and the release of uncontrolled cytokines and hormones, such as MCP-1, IL-6, IL-1β and TNF-α, but also leptin and adiponectin, by the adipose tissue [17,38]. In addition, the levels of other pro-inflammatory cytokines, IL-8, IL-1β and IFNγ are reported to increase in IR [39,40]. This inflammatory condition leads to the onset of obesity-related disorders, which are the result, at least in part, of changes in the secretion of adipose tissue-derived factors and explain the relation between inflammatory condition and metabolic response [7]. Chemokines are proinflammatory cytokines that stimulate leukocyte chemoattraction and are produced in response to infectious and other inflammatory stimuli by several different cell types [41]. Among pro-inflammatory chemokine, CXCL5, a chemokine ligand of CXCR2, has been related to insulin resistance and obesity-related disorders.

Indeed, the role of the CXCR2 pathway in obesity-induced inflammatory response was confirmed by the observation that a CXCR1/2 inhibitor was effective in the db/db mouse model in reducing neutrophil and macrophage infiltration in the liver and in the adipose tissue [42]. The observation that CXCR2 (−/−) mice are resistant to diet-induced insulin resistance and diabetes [22] prompted numerous studies aimed at clarifying the neutrophil-dependent and independent mechanisms implicated. CXCL5 neutralization was less effective than CXCR2 antagonism of receptor depletion in the knockout model, thus suggesting the involvement of other CXCR2 ligands that may contribute to the disease development in the mice model.

Recent data have shown that, along with CXCL5, circulating levels of CXCL1, with high affinity to CXCR2, are markedly increased in the db/db mice and correlate with obesity development [43,44]. KC, the human orthologue of CXCL1 (IL-8), is an important member of the chemokine family of pro-inflammatory chemotactic cytokines and it is also involved in systemic immunity and macrophages infiltration and activation in adipose tissue [45]. IL-8 was found increased in the blood of patients with T2D and its circulating levels positively correlated with worse inflammatory state and metabolic control [21]. Recent data highlighted a specific adipocytes phenotype in CXCR2 (−/−) mice that show a thin subcutaneous adipose layer associated with the small size of adipocytes [46], thus showing an unprecedented role of CXCR2 signalling on the expression of adipogenesis related genes.

Starting from these observations, in this paper we have focused our attention on KC and its specific role in the modulation of murine differentiated adipocytes response to stressed conditions. Our data support the concept that hyperglycemia strongly increases the expression of CXCR2 and its ligand CXCL1/KC. At a lower extent, exogenous stimulation with KC or cultured activated macrophage conditioned medium also showed the ability to induce the expression of CXCL1/KC. A similar trend was observed when looking at key mediators of the inflammatory pathways activated by adipocytes in response to HG and/or INF/KC showing a pronounced pro-inflammatory effect of HG leading to a significant increase in the expression of TNF-α, IL-6, NFkb and a corresponding reduction of the anti-inflammatory cytokine IL-10. Ladarixin, an allosteric inhibitor of CXCR1 and CXCR2, showed the ability to inhibit the overexpression of both CXCL1/KC and CXCR2 and to block the HG- or INF/KC-induced activation of all the analyzed pro-inflammatory signals, thus demonstrating a pivotal role of CXCR2 signaling in triggering the inflammatory cascade. These results were confirmed using proteome profiler arrays on a large panel of inflammatory cytokine mediators, showing the potential of the treatment in preventing the amplification of the inflammatory reaction in the AT driven by diabetes-associated stress and inflammation. Interestingly, these results were confirmed in primary human adipocytes derived from an obese patient thus reinforcing the translational significance of these findings. It is noteworthy that both in murine and human adipocytes, among the other inflammatory factors, a marked reduction of the secreted levels of CCL-2/MCP-1 has been observed upon ladarixin treatment. MCP-1 is a member of the chemokine superfamily that functions in the recruitment and activation of monocytes during inflammation, which stimulates obesity-associated macrophage infiltration, and its gene expression is sensitive to insulin levels. Hypertrophic adipocytes express MCP-1 receptor C-C motif chemokine receptor 2 [47].

In this study, we have also evaluated the impact of CXCR2 inhibition by ladarixin on relevant pathways of glucose metabolism and insulin resistance. In normoglycemic conditions, INF and KC stimuli did not affect glucose uptake or expression of the glucose transporters GLUT4 and GLUT1. Interestingly, ladarixin completely reverted GLUT4 and GLUT1 downregulation induced by HG and by INF/KC in combination with HG partially restoring glucose uptake in mature adipocytes. Coherently with the hypothesized role of CXCR2 on adipocyte insulin-sensitivity, a significant increase of adiponectin levels, markedly reduced in HG conditions w/o INF/KC stimuli, was observed.

It is noteworthy that ladarixin treatment was found to increase secreted leptin levels in murine adipocytes in all the tested conditions except for the HG plus INF/KC conditions. The behavior associated with CXCR2 inhibition in respect to leptin release is perfectly complementary to the trend observed for AKT and IRS-1/2 phosphorylation assay where an inhibitory effect in normoglycemic and HG conditions is converted to activation in HG plus INF/KC. The activation of the leptin pathway may explain the apparent discrepancy in the effect of CXCR2 inhibition on the insulin receptor activation pathway. While leptin controls glucose homeostasis by functioning as an insulin-sensitizing factor in most insulin target tissues, leptin has been reported, at high concentrations, to impair insulin-stimulated MAPK activity, glycogen synthase kinase 3 beta phosphorylation and insulin receptor tyrosine phosphorylation [48]. This is consistent with the effect on glucose uptake and transport that is conserved across the different conditions. In our experimental conditions, ladarixin effect on leptin levels, observed in murine adipocytes, is not conserved in human cells where a decrease of protein levels is shown.

Our molecular mechanisms investigations confirmed previous data obtained from another research group in muscle cells [22], showing the ability of CXCR2 inhibition to promote AKT and PI3K phosphorylation and insulin receptor substrates activation, key events in insulin-mediated signaling, confirming that ladarixin treatment may exert a dual beneficial action in the management of obesity-related disease by preventing the harmful pro-inflammatory signal activated by adipose tissue in response to inflammation and high glucose concentration and restoring the adipocyte functional insulin-sensitivity potentially preventing insulin resistance.

The relationship between the KC/CXCR1/2 axis and the development of insulitis and type 1 diabetes in mice has been previously established, demonstrating that transient CXCR1/2 inhibition by ladarixin prevents inflammation-mediated islet damage in multiple low-dose streptozotocin mice [24]. The compound is currently under phase III clinical investigation in the onset of type 1 diabetes to further evaluate whether ladarixin is effective in preserving β-cell function and slowing down the progression of the disease in type 1 diabetes patients.

The data collected in this work indicate a potentially beneficial effect for ladarixin in the management of obesity associated T2D. Ladarixin is a dual CXCR1/2 inhibitor cross-reacting with ortholog receptors in rodents. While the role of mouse CXCR1, considered for a long time a non-signaling receptor, and more recently deorphanized as a CXCL6 ligand, remains elusive, the role of the two receptors in humans is strictly cooperative and IL-8, the human orthologue of CXCL1/KC binds and activates the two receptor subtypes, both expressed on human adipocytes. A significant advantage associated with the allosteric mechanism of action is the ability of ladarixin to inhibit with comparable potency receptor activation mediated by different ligands thus overcoming the intrinsic redundancy of the chemokine signaling (i.e., CXCL1, CXCL5).

Ladarixin [25], was previously shown to prevent and reverse diabetes in non-obese diabetic (NOD) mice by selectively inhibiting pancreatic inflammation and preserving residual β-cell function [24]. A phase 3 multicentre, double-blind, placebo-controlled study with ladarixin is currently recruiting (ClinicalTrials.gov Id: NCT04628481) and aims to assess the efficacy of ladarixin in patients with new-onset type 1 diabetes.

The emerging evidence on the potential relevance of CXCR1/2 signaling in the onset and progression of obesity-driven metabolic diseases and related complications is investigated in phase 2 clinical studies in obese Type 2 Diabetes (started at December 2020, EudraCT Nr. 2020-003296-18) to assess the potential of Ladarixin treatment in the management of diabetes and obesity-related disorders.

## 5. Conclusions

This work reported the ability of CXCR2 inhibition through ladarixin treatment to exert a dual beneficial action in the cellular models tested (differentiated 3T3-L1 and primary human adipocytes derived from an obese patient), both preventing the harmful pro-inflammatory signal activated by adipose tissue in response to inflammation and high glucose (as observed in cytokine proteome array, NFKB and cytokines expressions) and restoring the adipocyte functional insulin-sensitivity (as observed in glucose transport analysis), potentially preventing insulin resistance mechanisms. On the light of the results obtained in this study, we can postulate that ladarixin could be a potential treatment in type 2 diabetes mellitus and obesity.

## Figures and Tables

**Figure 1 cells-10-02324-f001:**
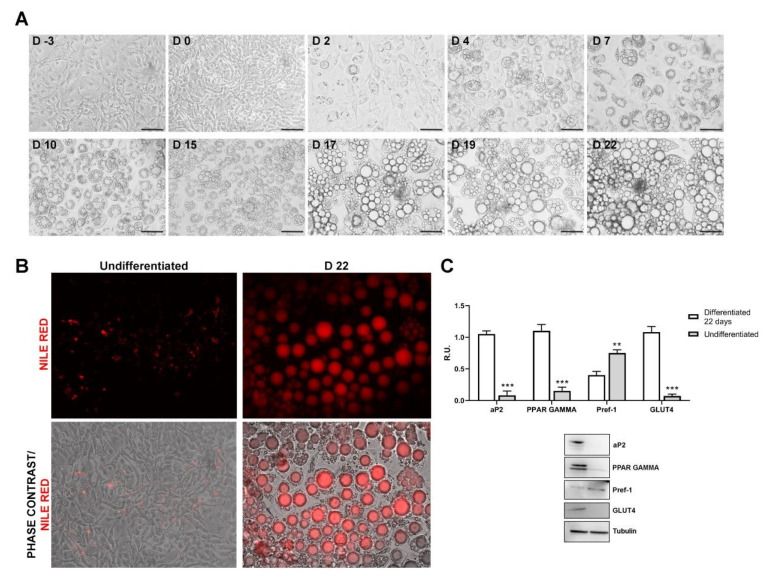
(**A**) Contrast phase images at different time points showing the adipocyte differentiation (Scale bar = 100 μm), (**B**) Confluent 3T3-L1 cells stained with Nile Red prior to induction and after 22 days of differentiation. (**C**) Western blotting and relative densitometric analyses of 3T3-L1 prior differentiation and after 22 days of differentiation. Data are mean ± SE of 3 different experiments. *t*-test ** *p* < 0.005, *** *p* < 0.0005 vs. differentiated cells (N = 3).

**Figure 2 cells-10-02324-f002:**
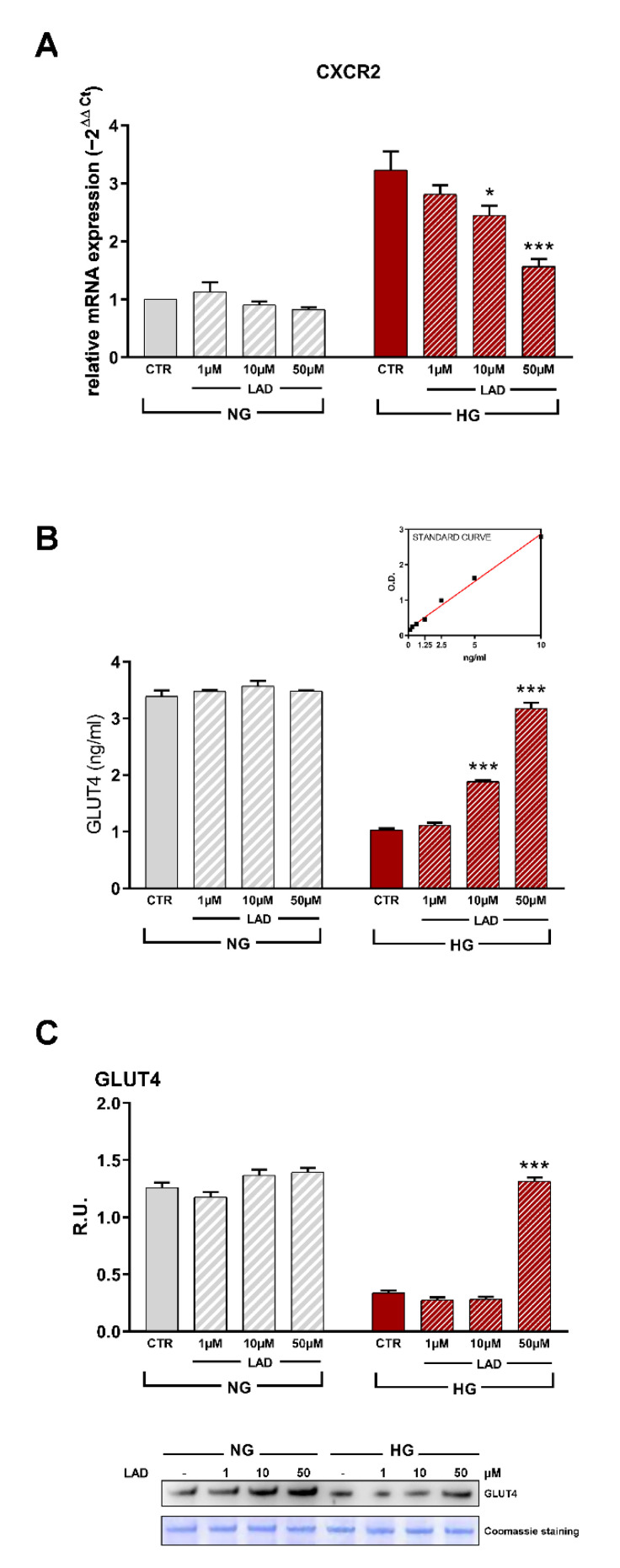
(**A**) RT-PCR for CXCR2. Data are mean ± SE of 3 different experiments; One way ANOVA, * *p* < 0.05, *** *p* < 0.0005 vs. CTR (N = 3). (**B**) ELISA kit analyzing GLUT4 protein levels in plasma membrane and (**C**) Western blotting with relative densitometric analyses. A representative band is shown. Data are mean ± SE of 3 different experiments; One way ANOVA, *** *p* < 0.0005 vs. CTR (N = 3).

**Figure 3 cells-10-02324-f003:**
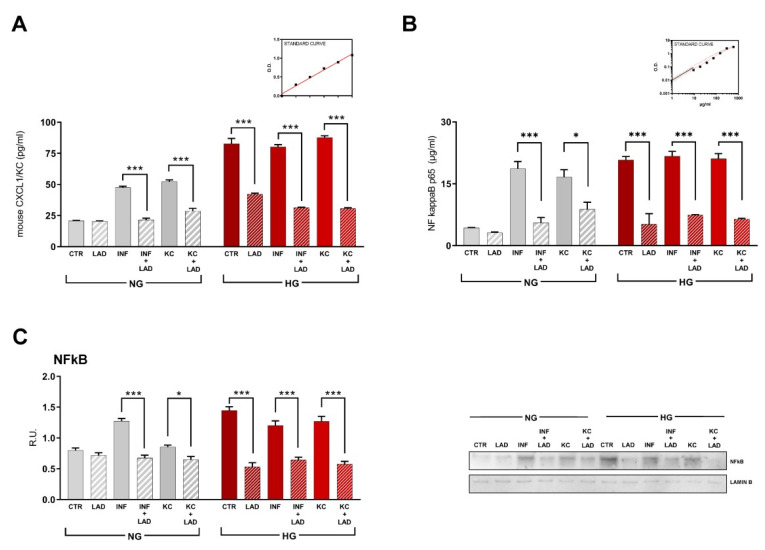
(**A**) KC (mouse IL-8) ELISA assay in mature adipocytes at different conditions. (**B**) Active form of NFkB (pS536) in mature adipocytes at different conditions analyzed by ELISA kit. (**C**) Western blotting of the nuclear fraction of NFkB and relative densitometric analyses. A representative figure is shown. Data are mean ± SE of 3 different experiments; One way ANOVA, * *p* < 0.05; *** *p* < 0.0005 vs. their respective condition without LAD treatment (N = 3).

**Figure 4 cells-10-02324-f004:**
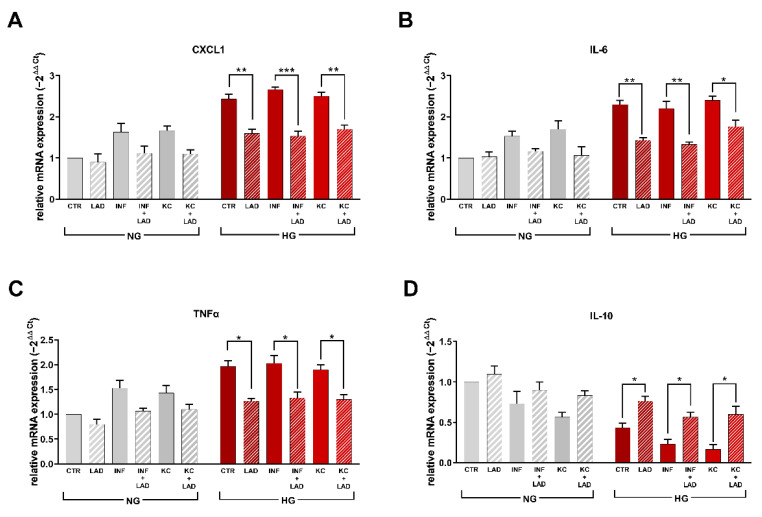
RT-PCR for Cxcl1 (**A**), Il-6 (**B**), Tnfα (**C**) and Il-10 (**D**) for mature adipocytes at different conditions. Data are mean ± SE of 3 different experiments; One way ANOVA, * *p* < 0.05; ** *p* < 0.005, *** *p* < 0.0005 vs. their respective condition without LAD treatment (N = 3).

**Figure 5 cells-10-02324-f005:**
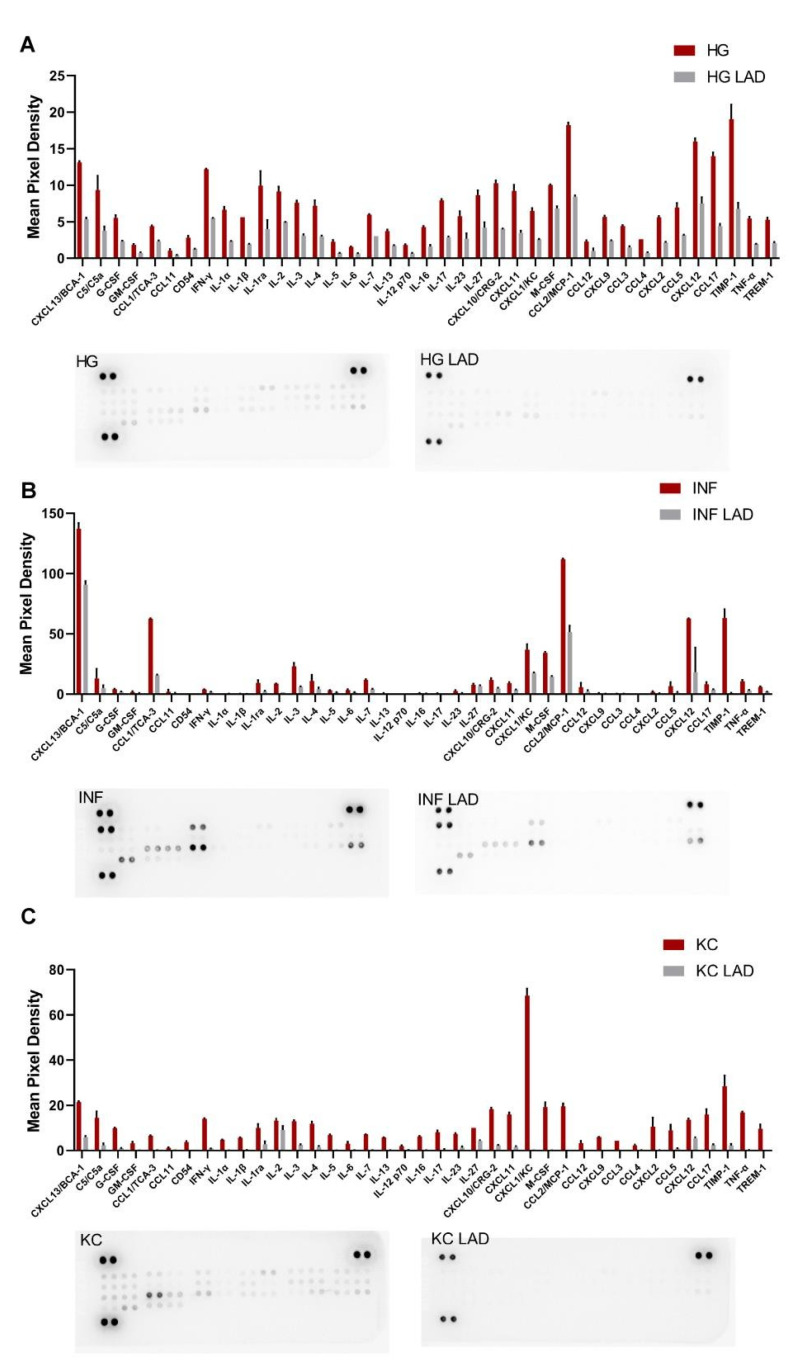
Cytokine production evaluated using proteome profiler arrays for mature adipocytes at different conditions: HG vs. HG+LAD (**A**), HG+INF vs. HG+INF+LAD (**B**), HG+KC vs. HG+KC+LAD (**C**). (N = 3).

**Figure 6 cells-10-02324-f006:**
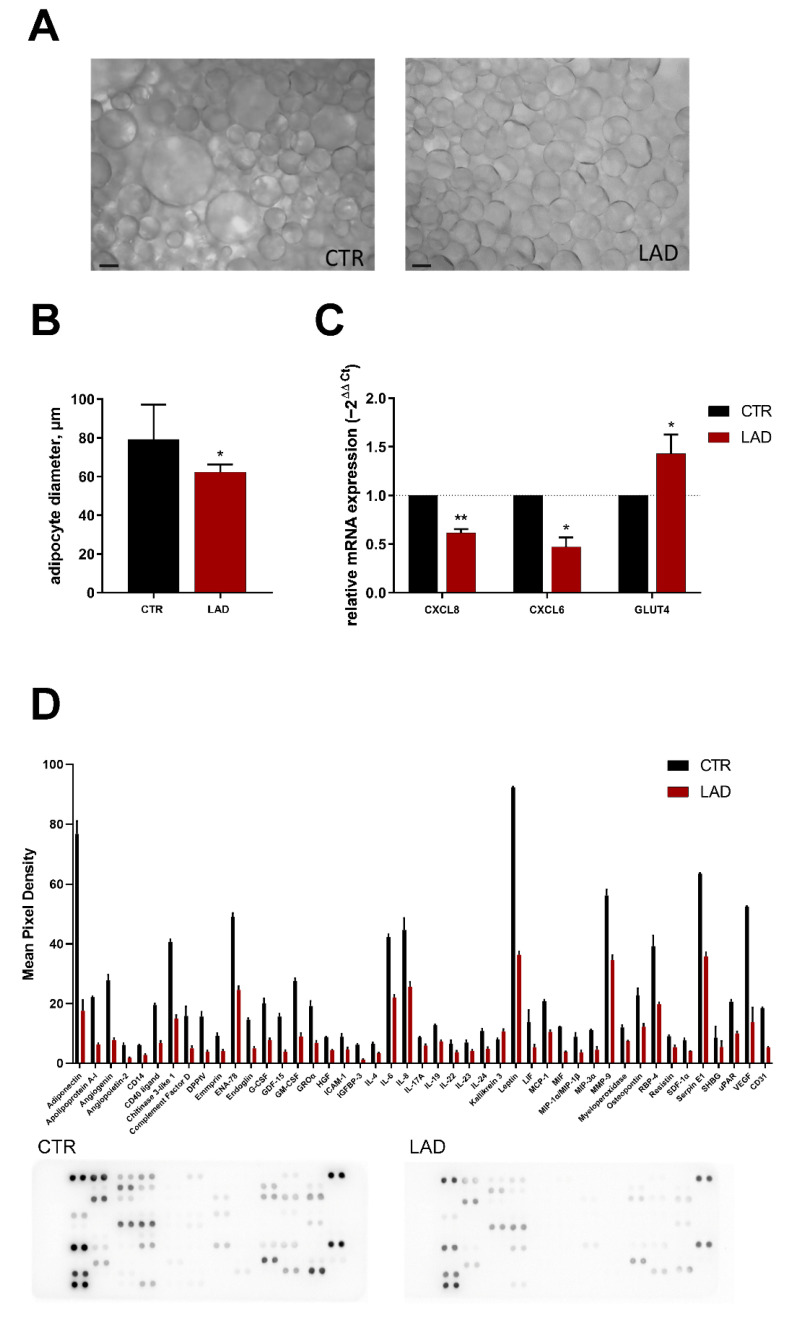
(**A**) Phase contrast representative figure of human adipocytes in CTR (obese) and LAD conditions. (**B**) Adipocyte size (µm) in CTR (obese patient, black) and LAD-treated (red). One way ANOVA, * *p* < 0.05 vs. CTR. Scale bar = 50 µm. (**C**) RT-PCR for CXCL8, CXCL6 expression and GLUT4 levels in human adipocytes of a 69-year-old obese woman in CTR and LAD treated conditions. One way ANOVA, * *p* < 0.05; ** *p* < 0.005, vs. CTR. (**D**) Human XL Cytokine Array for conditioned media of human adipocytes from an obese woman 69 years old (CTR) and treated with ladarixin for 72 h (LAD) (N = 3).

**Figure 7 cells-10-02324-f007:**
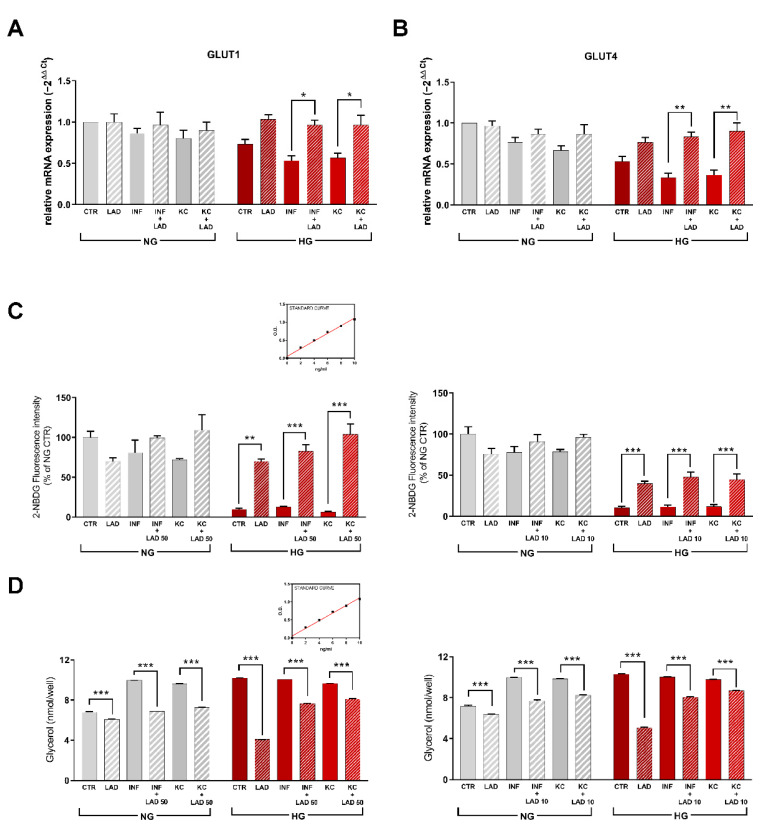
RT-PCR for GLUT1 (**A**) and GLUT4 (**B**) for mature adipocytes at different conditions. Data are mean ± SE of 3 different experiments; One way ANOVA, * *p* < 0.05; ** *p* < 0.005, vs. their respective condition without LAD treatment. (**C**) Glucose uptake in mature adipocytes at different conditions: on the left, 50 µM was the tested concentration while in the right 10 µM was the tested concentration. Data are mean ± SE of 3 different experiments; One way ANOVA, ** *p* < 0.005; *** *p* < 0.0005, vs. their respective condition without LAD treatment. (**D**) ELISA kit analyzing lipolysis in mature adipocytes at different conditions: on the left, 50 µM was the tested concentration while in the right 10 µM was the tested concentration. Data are mean ± SE of 3 different experiments; One way ANOVA, *** *p* < 0.0005, vs. their respective condition without LAD treatment (N = 3).

**Figure 8 cells-10-02324-f008:**
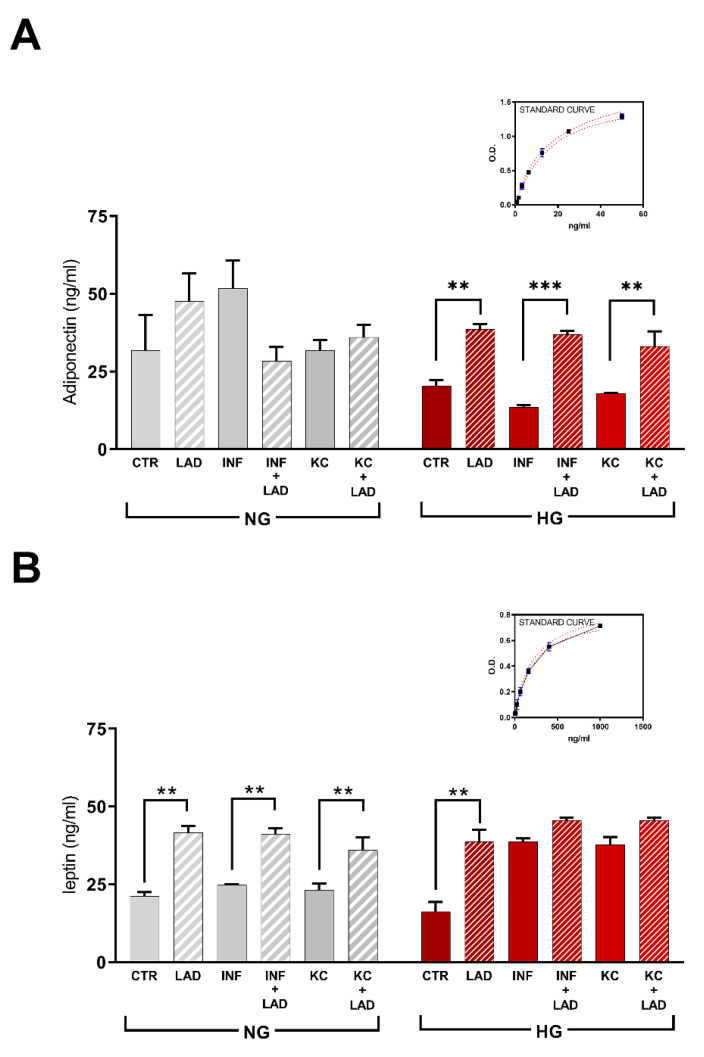
Adiponectin (**A**) and leptin (**B**) secretion in mature adipocytes at different conditions analyzed by ELISA kit. Data are mean ± SE of 3 different experiments; One way ANOVA, ** *p* < 0.005; *** *p* < 0.0005, vs. their respective condition without LAD treatment (N = 3).

**Figure 9 cells-10-02324-f009:**
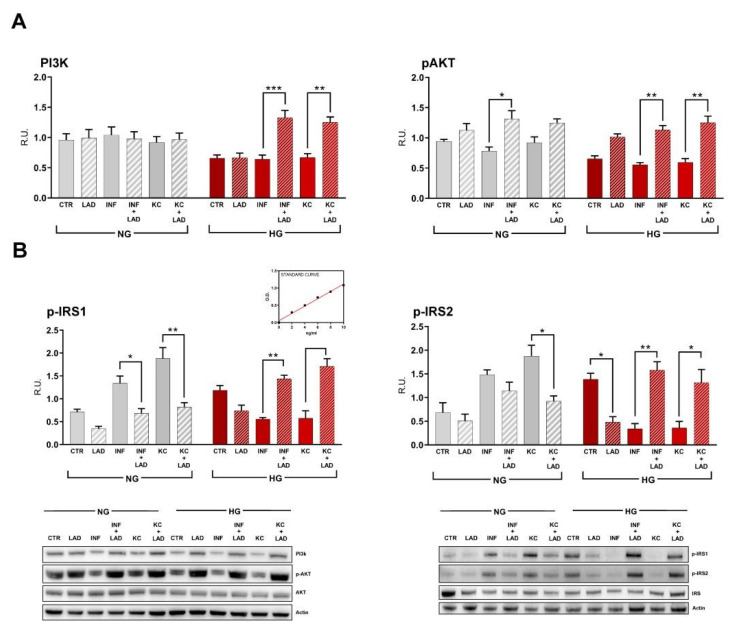
Western blotting analyses and relative densitometric analyses for PI3K and p-Akt (**A**) and p-IRS1 and p-IRS2 (**B**). A representative figure is shown. Data are mean ± SE of 3 different experiments; One way ANOVA, * *p* < 0.05; ** *p* < 0.005; *** *p* < 0.0005, vs. their respective condition without LAD treatment (N = 3).

## Data Availability

The datasets generated during the current study are available from the corresponding authors upon reasonable request.

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
