# Peer review of "CXCR1/2 Inhibitor Ladarixin Ameliorates the Insulin Resistance of 3T3-L1 Adipocytes by Inhibiting Inflammation and Improving Insulin Signaling"

_cells, 2021, doi:10.3390/cells10092324_

Round 1

Reviewer 1 Report

“CXCR1/2 inhibitor ladarixin ameliorates the insulin sensitivity of 3T3-L1 adipocytes by inhibiting inflammation and improving insulin signalling”, Castelli et al.

This manuscript describes the effect of ladaxin, a CXCR1/2 inhibitor, on the insulin sensitivity in adipocytes. The authors show that ladarixin acts positively on insulin sensitivity, glucose transporters GLUT1 and GLUT4, cytokine proteome profile, and lipid metabolism in adipocytes.

This is a very interesting study that suggests that ladarixin could be a potentially helpful treatment in type 2 diabetes mellitus and obesity.

The job has been well done and looks pretty convincing. However, some details are disturbing and require clarification.

First, the intro's "introduction" is far too long and takes up almost all the space, leaving only a few lines for the results and the conclusion. The authors should modify it and explain more fully the results obtained in the study.

In Fig. 1C, the authors show that the expression of certain markers of differentiation of adipocytes is increased during the differentiation of preadipocytes into adipocytes. It would be interesting to show the expression of GLUT4 under these conditions.

In Figure 2C, the authors isolated the plasma membrane of adipocytes. Could they blot the membranes with a recognized plasma membrane marker like alpha1 Na / K-ATPase (see Hajduch et al, Diabetologia 1998). That way, we would be sure that the plasma membranes are pure.

Line 509-510, the authors suggest that the size of human adipocytes decreases in response to ladarixin. It would be nice to seriously measure the size of the cells to be able to make such comparison.

In Figure 7, the authors measure glucose transport in adipocytes with or without ladarixin. They found that ladarixin increased glucose transport in adipocytes incubated in 50 mM glucose and inducers of inflammation. It would be interesting to compare this increase with that found in response to insulin. The authors should therefore also measure glucose transport in response to the hormone.

Some problems appear in figure 9. Indeed, the authors measure the PI3K. And they find that the latter's expression is increased in response to ladarixin (Figure 9A). However, the antibody they used is an antibody directed against the p110 subunit of PI3K. Is this representative of the activity of PI3K? This means that ladarixin induces an increase in the latter. The authors must say more about this fact.

In addition, the authors must specify against which residue the anti IRS1 / 2 antibodies are directed. Same for Akt ...

As in Figure 7, the authors should compare the effects of ladarixin with those of insulin. Does ladarixin restore the action of insulin which must be inhibited in these deleterious conditions?

Reviewer 2 Report

Here, Authors use CXCR1/R2 antagonist, ladarixin to characterize its effect on inflammation/high glucose-induced insulin resistance in adipocytes. The study is very interesting but can be improve by addressing following points. Title of the manuscript is misleading/confusing and not supported by conclusion. probably it should be "ameliorates the insulin resistant" rather than "ameliorates the insulin sensitivity" Otherwise this in vitro study is well conducted and conclusion is well supported by experimental results

Round 2

Reviewer 1 Report

The authors have greatly improved their manuscript. They answered the majority of my questions when it was possible to do so in the allowed time. Nevertheless, the authors should include the details about the phospho-antibodies they use in Figure 9 in the legend of this figure. Other than this detail, I think the manuscript can be published in Cells.